# The Role of Diet and Interventions on Multiple Sclerosis: A Review

**DOI:** 10.3390/nu14061150

**Published:** 2022-03-09

**Authors:** Panagiotis Stoiloudis, Evangelia Kesidou, Christos Bakirtzis, Styliani-Aggeliki Sintila, Natalia Konstantinidou, Marina Boziki, Nikolaos Grigoriadis

**Affiliations:** 2nd Department of Neurology, AHEPA Hospital, Faculty of Health Sciences, Aristotle University of Thessaloniki, 54636 Thessaloniki, Greece; stpan@windowslive.com (P.S.); bioevangelia@yahoo.gr (E.K.); bakirtzischristos@yahoo.gr (C.B.); lina17puma@yahoo.gr (S.-A.S.); nataliak95@gmail.com (N.K.); bozikim@auth.gr (M.B.)

**Keywords:** multiple sclerosis, diet, nutrition, gut–brain axis, gut microbiota

## Abstract

Multiple sclerosis (MS) is a chronic autoimmune disease of the central nervous system (CNS) characterized by inflammation and neurodegeneration. The most prominent clinical features include visual loss and sensorimotor symptoms and mainly affects those of young age. Some of the factors affecting its pathogenesis are genetic and/or environmental including viruses, smoking, obesity, and nutrition. Current research provides evidence that diet may influence MS onset, course, and quality of life of the patients. In this review, we address the role of nutrition on MS pathogenesis as well as dietary interventions that show promising beneficial results with respect to MS activity and progression. Investigation with large prospective clinical studies is required in order to thoroughly evaluate the role of diet in MS.

## 1. Introduction

Multiple sclerosis (MS) is a chronic autoimmune disease of the central nervous system (CNS) characterized by loss of myelin and inflammation, leading to neurodegeneration. Clinical features mainly include visual loss and sensorimotor symptoms as well as more atypical features such as fatigue and mental/cognitive impairment. It affects mainly patients of young age and mostly women [1]. It is classified in three clinical forms: relapsing-remitting (RRMS), primary progressive (PPMS), and secondary progressive (SPMS) disease [2], characterized by varying degree of pathology across the spectrum of acute/chronic inflammation and/or neurodegeneration. Its prevalence varies, with Europe and North America reporting the highest prevalence. Diagnosis is based on the revised 2017 McDonald criteria [3]. For RRMS, a clinical attack and dissemination in time and space is required. For PPMS, there is a need for disability progression confirmed for at least one year and dissemination in space. The SPMS subtype requires disability progression following the initial diagnosis of RRMS [3]. Several disease modifying treatments (DMTs) are currently in use in order to manage the ongoing disease activity in an attempt to control relapses and disability progression [4].

The pathogenesis of MS remains complicated and multifactorial. Other than genetic, various environmental factors seem to play a role in the development of MS. Microbial and viral infections, smoking, vitamin D, sun exposure, obesity, and dietary habits may be relevant to its pathogenesis. Environmental factors not only affect the development of MS but also the disease course and progression. Conversely, physical exercise and healthy diet appear to have an anti-inflammatory effect and to, at least in part, ameliorate the disease course [1,5,6,7,8].

Nutrition and dietary factors affect the mechanisms of MS pathology, its development, and degree of activity [9]. Although studies have shown the important role of nutrition in MS, the current therapy is not combined with any specific nutritional or lifestyle recommendation [1].

The present article reviews the current literature concerning the association of nutrition with the pathogenesis of MS and dietary interventions that affect its course.

## 2. Mechanisms of MS Pathology—The Effect of Diet/Nutritional Factors

### 2.1. Neurodegeneration

It is already known that neurodegeneration is presented even at the earliest stages of the disease [5]. In experimental models, oxidative stress leads to mitochondrial dysfunction, causing cell membrane disruption and eventually neuronal cell death [9]. Dietary antioxidant factors can dampen oxidative stress and may help against chronic demyelination and neuronal or axonal damage [5]. Both oxidative and mitochondrial injury primarily disrupt the function of neurons and glia, causing disturbances in cellular communication [10].

#### 2.1.1. Oxidative Stress

Oxidative injury is involved in both relapsing-remitting and progressive forms of MS [11]. Inflammatory cytokines, reactive oxygen species, and phagocytes lead to damage of myelin and axons. It is found that oxidative stress enhances inflammation and causes damage of the myelin, consequently leading to cell death. Clinically, the course of MS has been associated with inflammatory and oxidative stress mediators including cytokines such as IL-1β, IL-6, IL-17, TNF-α, and INF-γ [12].

Dietary antioxidant factors may regulate the activation of immune inflammatory cells, leading to the reduction in inflammatory and may also dampen oxidative stress, thus preventing chronic demyelination and axonal damage. Antioxidant factors such as curcumin, vitamin D, and fatty acids have been studied and seem to play a role in the regulation of oxidative stress [13]. Curcumin, derived from the plant *Curcuma longa* [12], has been advocated to inhibit proinflammatory cytokines [14]. In animal models of MS, curcumin was shown to reduce clinical severity and decrease CNS infiltration by inflammatory cells in mice. Curcumin possesses antioxidant and anti-inflammatory properties. Its anti-oxidant effects have been assessed in several neurodegenerative diseases including Alzheimer’s disease (AD), Parkinson’s disease (PD), and MS [15]. Another nutritional factor is melatonin, which is produced naturally by the pineal gland during the night. It is formed exogenously from tryptophan. Melatonin is mainly consumed from meat, oily fish such as salmon, eggs, milk, seeds, nuts, almonds, and soy products. Melatonin is suggested to regulate anti-oxidative defensive systems by stimulating the synthesis of superoxide dismutase and glutathione peroxidase, especially in patients with SPMS [16].

Vitamin D plays a significant role not only in calcium homeostasis and bone health, but also in immunomodulation and the reduction in oxidative stress. MS patients frequently exhibit vitamin D deficiency [1]. Studies report that low levels of vitamin D are associated with a higher risk for the development and relapse of MS [8,17]. Supplementation with vitamin D has been shown to have anti-inflammatory and immunomodulatory effects on MS pathogenetic mechanisms by inhibiting the production of CD4+ T cells, thus lowering the risk of MS and diminishing disease progression [18]. However, Bagur et al. reported in their systematic review that existing studies on the effect of vitamin D supplementation in MS are inconsistent with respect to EDSS, MRI lesions, overall functional status, and relapse rate [13]. It has been suggested that empirical replacement with high doses of vitamin D supplementation (at least 4000 IU/day orally) and for a prolonged period appears to be safe and is associated with low risk for adverse events, although available data are limited [12,19,20,21].

Vitamin A is a fat-soluble nutrient with a variety of functions in visual ability, skin, and immunity. Vitamin A includes retinoids and carotenoids, available in liver, milk, cheese, green leaves, oil, vegetables, and fruit. Association between the pathogenesis of MS and vitamin A remains undefined. Studies in animal models demonstrate a possible role of vitamin A in the modulation of immunity [22,23]. A negative correlation has been found between the development of MS and low levels of vitamin A in plasma [12]. A randomized controlled trial showed benefits in fatigue, depression, and cognitive status of MS patients supplemented with high doses of vitamin A (400 IU/day), which were considered safe and were not associated with adverse effects [24].

Fatty acids, especially omega-3 polyunsaturated fatty acids (PUFAs), are other antioxidant compounds that are associated with ameliorating neurodegeneration in MS. Intake of PUFAs consumed via fish, nuts, and seeds seems to be associated with protective effects against demyelination [5]. In animal models, PUFAs decrease inflammation, maintain immunomodulation and promote neuroprotection and remyelination [5]. Some studies have shown inconsistent results indicating the effect of PUFAs mainly against progression. In one study, association between PUFA intake and MS incidence seems to be non-significant. Conversely, one Swedish and one Australian study reported low incidence of MS in people following diets enriched in PUFAs [5,12,13,25,26]. Results from meta-analyses suggest that PUFAs may reduce the frequency of relapses, but are not effective against the progression of the disease [1,19]. In human studies, a low fat diet supplemented with PUFAs was associated with lower levels of disability assessed by EDSS, slight improvement in relapse rat, as well as improved quality of life [13,25]. Another study provided evidence of PUFA-related improvement with respect to specific markers linked with inflammation and/or neurodegeneration in patients with MS (for instance, matrix metallopeptidase-9 (MMP-9) rather than in quality of life, EDSS score, or fatigue [26].

Among PUFAs, α-linolenic acid (ALA) is associated with low incidence of MS. It can contribute to the immune pathway by decreasing markers of inflammation. Eicosapentaenoic acids (EPAs) and docosahexaenoic acids (DHAs) can also play a role in in decreasing MMP-9 levels in patients with MS [25]. Riccio et al. reported that fish oil supplementation enriched with omega-3 fatty acids have a beneficial effect in the inhibition of the expression and reduction in the levels of MMP-9 in MS patients [27]. Ramirez et al. reported the beneficial effects of fish oil containing high amounts of omega-3 PUFAs into protecting against inflammation and oxidative stress [25]. Omega-3 fatty acid supplementation results in the decrease in proinflammatory cytokines, free radicals, and as a result, improving the quality of life of patients with MS by decreasing relapse rates [25].

Polyphenols, which are included in vegetables, fruit, wine, and tea, have been proven to be beneficial, leading to modulation of the immune response and affecting the expression of genes encoding pro-survival proteins including antioxidant enzymes. Polyphenols can also enhance neuronal survival [28]. Studies have focused particularly on polyphenols such as resveratrol and ginkgo biloba. In animal studies, these compounds seemed to promote protection against oxidative stress, also protecting against demyelination and axonal injury [26]. Khalili et al. suggested that lipoic acid consumption by patients with MS results in the improvement of total antioxidant capacity [13].

Randomized clinical trials seem to confirm the efficacy of some of the compounds discussed above such as melatonin, vitamin D3, omega-3 PUFAs, and polyphenol compounds. However, further research is needed in order to understand the potential protective effects exerted by antioxidants on the cellular immunology of MS neurodegeneration [12].

#### 2.1.2. Mitochondria—Energy Production

Mitochondrial injury or the accumulation of iron in the brain is also enhanced in the progressive phase of the disease [12]. In patients with MS, mitochondrial structural changes and enzyme activity increase ROS production and cause oxidative damage [12]. Among the other antioxidants, curcumin is especially reported to play a major role against free radicals. Curcumin may benefit patients with MS by binding transition metals and forming stable inactive complexes, especially with ferrous ions, protecting against neurodegeneration [29].

### 2.2. Immune System (Innate and Adaptive) Responses—Factors of Immune System Activation

Nutrients and special diets such as saturated and ‘trans’ fatty acids, α-lipoic acid, polyphenols, high-fat diet, and high-carbohydrate diet result in the modulation of the components of inflammatory cascade. Several studies have shown that saturated and ‘trans’ fatty acids and lipopolysaccharide (LPS) may upregulate the activity of proinflammatory compounds, promoting inflammation; on the other hand, calorie restriction, polyphenols, and Ω-3 PUFAs would exert the opposite effect [26]. The influence of diet on inflammatory and autoimmune processes in MS is highlighted, supporting the hypothesis of a close relationship between nutritional factors and the immune system responses that play a role in the pathogenesis of MS [26].

### 2.3. Proinflammatory Diet

Recent studies have highlighted the role of proinflammatory diets in the pathogenesis of MS. Fatty acids and polyphenols as well as diets high in carbohydrates and fats may induce inflammatory cascade [26]. Diet can induce the production of inflammatory factors such as tumor necrosis factor, interleukins, MMP9, prostaglandins, and leukotrienes, leading to inflammation and oxidative stress [26].

Swank et al. reported adverse effects of saturated fatty acids (SFAs) on the course of MS, emphasizing their proinflammatory character [9]. High intake of SFAs leads to a dysbiosis of gut microbiota. Additionally, the consumption of vegetable oils, which are enriched with trans fatty acids, is associated with gut inflammation and the upregulation of proinflammatory cells [30]. Red meat leads to the formation of nitrous compounds increasing chronic inflammation. Red meat also contains arachidonic acid, which participates in inflammatory pathways by activating Th17 cells [27]. Furthermore, a high consumption of sugar-sweetened beverages and refined cereals leads to the production of insulin, which, in this way, is responsible for the upregulation of synthesis and the production of arachidonic acid. High salt intake can induce the production of Th17 cells and proinflammatory cytokines [27]. Proteins contained in cow-milk may play a role in the mechanisms of pathogenesis of MS. Particularly, butyrophilin can induce EAE by mechanisms of molecular mimicry with myelin oligodendrocyte glycoprotein [27].

### 2.4. Gut Brain-Axis and MS

#### 2.4.1. Gut Microbiota

The gut–brain axis represents a bidirectional communication system between the CNS and the gastrointestinal system that includes the CNS, the enteric nervous system, the autonomic nervous system, the immune system, and the gut microbiota [31,32]. The role of gut microbiota is crucial because of its impact on regulating and maintaining the normal function of the innate immune system [31,32]. From birth to adolescence, commensal microbiota infests the gastrointestinal system, remaining in a stable condition, a state called eubiosis [31,32]. However, in early stages of life, factors such as antibiotics, infections, or unhealthy dietary habits may lead to alterations of the relative distribution and frequency of commensal microbiota, thus also, at least in part, predisposing to gut colonization by pathogens, a state called dysbiosis. In dysbiosis, there is an increase in the number of pathogenic bacteria and decrease in their biodiversity, resulting in gastrointestinal and systemic inflammation, possibly leading to increased risk for local or systemic inflammatory disease [31,32].

Studies on the experimental model of MS show the possible association of gut microbiota with the severity of the disease, indicating a possible protective role as well as a role in inducing pathological mechanisms in the context of immune dysregulation in CNS autoimmunity. The presence of gut commensal microbiota is necessary for the occurrence of CNS autoimmunity [33]. In animal models, it has been shown that the gut is involved in the modulation of inflammation of the CNS and may orchestrate mechanisms of immune tolerance, thus protecting from the development of CNS autoimmunity [34]. Metagenomic studies addressing the gut microbiota composition by next generation sequencing (NGS) demonstrated microbial imbalance and differences in the relative composition of gut microbiota in MS patients compared to healthy individuals, linking the dysbiosis with possible MS pathogenesis [35,36]. In this respect, nutritional modification with a potential to modulate the gut commensal microbiota has been advocated as a strategy that may affect the development of the disease and/or alter the disease course [1,32,33,37,38].

Metabolism of nutrients, especially carbohydrates, the production of neurotransmitters and vitamins, and competition with other colonizing pathogens are some of the main physiological functions of gut microbiota [32]. Furthermore, gut microbiota is possibly associated with CNS homeostasis and development and also with neuroimmunological and neurodegenerative disease [32]. Diet comprises a main factor determining the synthesis and metabolism of gut microbiota, thus enabling the host to defend against pathogens. The role of gut microbiota is also significant for the regulation of the immune system by affecting the overall activation status of T cells and other cells of the innate and adaptive immunity. In particular, T regulatory cells and T helper cells type 2 may suppress the activation of the immune system [1,32,33,39]. Moreover, short-chain fatty acids (SCFAs) such as butyrate, derived from gut microbiota, promote anti-inflammatory processes by producing anti-inflammatory cytokines and by inhibiting the connection of leukocytes to epithelium [1,32,33,39]. It is believed that the consumption of a diet with high fiber intake may increase the production of butyrate, thus leading to improved outcomes in patients with CNS disorders [1,32,33,39]. Studies in animal models demonstrated a strong and important connection between the microbiota, butyrate production, and the CNS. Patients with MS have lower levels of SCFAs in feces as well as reduced frequency of SCFA-producing bacteria in the gut [1,32,33,39].

Nutrition and dietary interventions regulate the gut microbiota affecting its composition and its functionality [32]. Diets characterized by a high intake of fat, sugar, and animal protein may lead to the development of specific pathogenic bacteria species such as Bacteroidetes in the gut, which, in turn, may induce enteric inflammation, damage of the intestinal barrier and increase in cross-reactive cells of the adaptive immunity [31]. Moreover, diet-induced low biodiversity of gut microbiota is associated with metabolic changes as well as an increase in inflammation markers [31].

It has been proposed that a diet enriched with vegetables, a high amount of fiber combined with probiotics, vitamin D and vitamin A supplementation, and lipoic acid results in gut eubiosis. This leads to an increase in microbial diversity and microbe-associated anti-inflammatory mediators such as short chained fatty acids (SCFAs). Conversely, a diet rich in animal fat and trans-fatty acids also including sugar and salt intake, promotes gut dysbiosis and results in an increase in the presence of pro-inflammatory mediators and also in gut barrier and blood–brain barrier (BBB) permeability, resulting in CNS autoimmunity [40].

#### 2.4.2. Effects of Pre- and Probiotics in Patients with Multiple Sclerosis

The impact of diet on gut microbiota has been experimentally studied through the observations of effects of pre- and probiotics in patients with autoimmune diseases [37]. Prebiotics are nonviable compounds of living microorganisms with an ability to beneficially manipulate the host’s microbiota. Many fermentable carbohydrates have prebiotic effects. Non digestible oligosaccharides such as fructans and glycans, which are utilized by Bifidobacteria, are reported to have the most beneficial effects. In addition, oligasaccharides identified in dairy products are reported to act as prebiotics. Probiotics are mostly consumable live microorganisms such as Lactobacillus. Sources of probiotics are contained in food such as yogurt [38]. Kouchaki et al. reported improvement in EDSS scale and decrease in inflammatory markers in patients with MS who were treated with probiotic supplementation [39]. A number of studies have encouraged the use of pre- and probiotics in patients with MS due to their benefits in maintaining the homeostasis of the CNS, improving the intestinal microbial balance and regulating the composition of gut microbiota. In these studies, the combination of prebiotics and probiotics is highly recommended [27].

## 3. Comorbidities in MS as an Independent Factor of Pathology—The Effect of Diet and Nutrition

Results from recent studies have shown that the presence of factors that predispose toward cardio-vascular risk in MS patients not only increases the risk for higher disability in the context of MS, but is also associated with diagnostic delay in MS. In addition, vascular comorbidity is reported to be associated with higher risk of hospitalization for patients with MS [5]. Of these, hyperlipidemia and obesity are the most common comorbidities among patients with MS [5].

### 3.1. Hyperlipidemia

Hyperlipidemia has been reported to be a common comorbidity among patients with MS. The mechanism by which hyperlipidemia affects MS is currently unexplained, but it may involve fatty acid metabolic and inflammatory pathways that influence the regulation of gene expression and metabolism. Lipid composition of myelin and its morphology are affected during neuroinflammation. A high-fat diet is thought to promote neuroinflammation. Cholesterol and its components are associated with adverse disease effects on patients with MS. Moreover, it is hypothesized that cholesterol and its molecules could be markers of disease activity, markers of efficacy of treatment, or possible therapeutic targets [1]. It is also suggested that fatty acids are not only associated with neuroinflammation, but also with neurodegeneration, affecting the progression of MS [1]. High consumption of fatty acids, especially saturated fats, evidently leads to increased levels of blood LDL cholesterol. In particular, long chain fatty acids included in processed food influence the immune system by activating proinflammatory components, leading to T cell and macrophage activation and expression of inflammatory cytokines [5]. Swank et al. reported in an epidemiological study in patients with MS that low intake of saturated fat was associated with lower disability and mortality rate. However, no controlled randomized trials that have focused on saturated fat consumption exist, thus conclusions are difficult to make [5].

### 3.2. Obesity and Increased BMI

Patients with MS often consume a low-carbohydrate and high-lipid diet associated with abdominal obesity and higher body mass index (BMI). This condition leads to a pro-inflammatory status increasing levels of interleukin 6 (IL-6), TNF-alpha, and leptin, factors that are associated with MS pathogenesis [41]. Additionally, recent studies suggest that increased BMI and obesity play a major role in MS development and progression [42]. Obesity and MS can lead to altered adipokine release into the blood circulation. This activates inflammatory pathways and increases the infiltration of immune cells in the CNS. Moreover, increased levels of pro-inflammatory cytokines in MS release pro-inflammatory adipokines and disrupt adipokine pathway, creating a feedback loop [42]. Pathophysiologically, obesity affects MS by promoting chronic inflammation, altering the endocrine system by disturbing the secretion of adipokines and influencing the gut microbiota [42]. Moreover, an unbalanced diet contributes to an increase in BMI and abdominal obesity. BMI is affected by protein and lipid intake as well as carbohydrate intake. Excessive simple carbohydrate intake is related to obesity and being overweight as well as an increase in adipose tissue in the abdomen [43]. It has been reported that in MS patients, intake of lipid and protein favors abdominal obesity and increases BMI. This fact is associated with an increase in proinflammatory cytokines, promoting inflammation [43].

Overweight and obesity are associated with chronic inflammation of adipose tissue, which is accompanied by an altered secretion of adipokines. These adipokines are hormones and cytokines that regulate metabolic pathways [42]. Among others, leptin and adiponectin seem to be associated with the pathogenesis of MS. Leptin regulates energy and acute phase reactions, interfering in the activity and progression of experimental autoimmune encephalomyelitis (EAE) in mice or the pathogenesis of MS in humans [26]. Increased leptin in humans has been suggested to be related with proinflammatory conditions and autoimmunity. As a result, a diet associated with increased levels of leptin could possibly influence the balance between T cells, promoting inflammation and inducing cell-mediated autoimmunity [26]. In humans, hyperleptinemia has been correlated with pro-inflammatory conditions [44]. In patients with MS, expression of leptin receptors has been found to be significantly higher in the relapse phase than that observed in remission [45]. Piccio et al. suggested that caloric restriction, leading to a reduction in leptin levels, can reduce inflammation, demyelination, and axonal injury [46]. Adiponectin (APN) is another adipokine that exhibits anti-inflammatory activity on immune system cells [47]. Musabak et al. found that in patients with MS, the APN serum levels were lower than those of the healthy controls [48]. Their study found that low APN levels were associated with high risk for early onset of MS, especially in females, and also with increased disability and progression, predicted by higher EDSS score [48].

## 4. Dietary Patterns—Interventions and Effects on MS

Dietary approaches have been studied in order to improve outcomes in MS. However, several studies have provided at least some indications on the potential role of dietary habits in the course of MS. Among the most popular dietary interventions overall are the Mediterranean, the Paleolithic, the Swank, the McDougall, and the Hyberbolic-caloric restriction diets [1,31].

### 4.1. Mediterranean Diet

The Mediterranean diet consists of a high intake of fruit, vegetables, and whole grains including olive oil as the main source. It also includes a moderate amount of fish and dairy products and a low intake of red meat [5]. Phenols in olive oil are responsible for its anti-inflammatory effects, thus protecting the nervous system from oxidative stress. It seems that the Mediterranean diet reduces inflammatory markers [31], regulates predisposing factors of vascular pathology in the context of several autoimmune disorders [28,49], and regulates gut microbiota [28,49]. Thus, it has been suggested that the Mediterranean diet is associated with low risk for MS onset [5,30,32].

### 4.2. Paleolithic Diet

The Paleolithic diet is characterized by the consumption of leaf green vegetables, plant proteins, soy, and nuts excluding the consumption of dairy and processed food. Studies indicate that fatigue in MS patients following the Paleolithic diet was improved, although risk for nutritional deficiencies was increased [5,32].

### 4.3. Swank Diet

The Swank diet is based on limited saturated fat intake. Increased fruit, vegetable, and oil intake is encouraged. Swank observed that the risk of MS development was higher in people living in areas with a high consumption of fat [9]. This study showed that patients with MS who followed the Swank diet exhibited a lower risk for relapse, disease progression, and reduced mortality. Although possible underlying mechanisms have not been identified, it has been advocated that reduction in fat consumption may be related to protection against inflammation and demyelination [1,32].

### 4.4. McDougall Diet

The main caloric source of the McDougall diet is carbohydrates, based on consuming plants [5]. In addition, olive oil and animal products including eggs and dairy products are not preferred and restricted. Studies showed an association with lower fatigue in the group of MS patients who followed the McDougall diet [9,32]. Conversely, there was no significant effect on relapse rate, magnetic resonance imaging activity and disability [5,32].

### 4.5. Hyberbolic Diet-Caloric Restriction

It is believed that an increased amount of consumed calories is associated with inflammation, especially following meals [5,9]. As a result, caloric restriction has been suggested to reduce the risk of postprandial inflammation. Results from studies concerning caloric restriction revealed a reduction in oxidative stress in patients with relapsing and progressive types of MS, leading to a better quality of life [32]. Trials on the possible beneficial effect of intermittent fasting on relapse rate and progression of MS are currently ongoing [49,50]. Overall, following an anti-inflammatory diet, in terms of intermittent fasting, seems to regulate inflammation and protect against oxidative damage and progression of MS in experimental models [27]. Choi et al. reported in their study that an anti-inflammatory diet that included mainly caloric restriction diet showed promising results by reducing inflammation and promoting regeneration in animal models with experimented autoimmune encephalomyelitis [51].

### 4.6. Ketogenic Diet

The ketogenic diet is low in carbohydrates and high in fat [32]. It is characterized by the induction of ketones released in blood circulation that may exert anti-inflammatory effects [32]. Trials in MS patients revealed a possible improvement in quality of life, fatigue, and depression related to the ketogenic diet, although negative effects including deficiency of vitamins, weight loss, and gastrointestinal symptoms may also arise [49].

### 4.7. Gluten Free Diet

Current evidence does not show a significant effect of the gluten free diet in MS [49,50]. Few gluten-free interventions showed an improvement in EDSS, lesion activity, fatigue, and quality of life, however, existing studies pose limitations in terms of the high risk of bias and also the lack of controlled randomized trials [49,50].

## 5. Conclusions

Although a balanced diet involving high amount of fruit, vegetables, and low fat may not replace DMTs in controlling disease activity in MS, it may have an add on value in a more efficient management of the disease overall. Existing evidence indicates that nutrition and diet may play a role in MS pathogenesis and course. These factors may affect gut microbiota function, enzyme activity, and risk factors of vascular pathology in MS patients. At the moment, precise recommendations regarding a specific dietary plan in patients with MS do not exist. However, clinical and experimental studies provide indirect evidence that a balanced diet in combination with an overall healthy lifestyle is linked with an improvement in several clinical parameters as well as measurements of quality of life for patients with MS. Furthermore, we strongly suggest large, well-scheduled clinical trials containing both clinical and biochemical, molecular, metagenomic, and metabolomic technologies aiming to clarify the role of diet in MS management.

## Data Availability

Not applicable.

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
