# Peer review of "The Role of Diet and Interventions on Multiple Sclerosis: A Review"

_nutrients, 2022, doi:10.3390/nu14061150_

Round 1

Reviewer 1 Report

This is a clearly written and well-organized work which reviews the role of nutrition as well as the effects of different dietary interventions on multiple sclerosis pathogenesis. This review can be relevant and important for the audience to read. I would suggest to include a paragraph about the effects of pre- and probiotics as a diet supplement for multiple sclerosis patients.

Author Response

Dear Reviewer,

On behalf of all co-authors, we are grateful for your fruitful comments upon our manuscript entitled “The role of diet and interventions on multiple sclerosis: A review”

We have the honor to submit our revised manuscript based on the remarks that have been proposed.

Sincerely yours

N. Grigoriadis

Q1. This is a clearly written and well-organized work which reviews the role of nutrition as well as the effects of different dietary interventions on multiple sclerosis pathogenesis. This review can be relevant and important for the audience to read.

I would suggest to include a paragraph about the effects of pre- and probiotics as a diet supplement for multiple sclerosis patients.

We thank Reviewer#1 for his/her comment. We have now included a paragraph about the effects of pre- and probiotics as a dietary supplement in MS patients, §23, lines 210-221:

2.4.2 Effects of pre- and prebiotics in patients with multiple sclerosis

The impact of diet on gut microbiota has been experimentally studied through the observations of effects of pre- and probiotics in patients with autoimmune diseases [42]. Prebiotics are nonviable compounds of living microorganisms with an ability to manipulate beneficially the host’s microbiota. Many fermentable carbohydates have prebiotic effects. Non digestible oligosaccharides such as fructans and glycans which are utilized by Bifidobacteria are reported to have the most beneficial effects. In addition, oligasaccharides identified in dairy products are reported to act as prebiotics. Probiotics are mostly consumable live microorganisms like Lactobacillus. Sources of probiotics are contained in food such as yoghurt  [43]. Kouchaki et al reported improvement in EDSS scale and decrease in inflammatory markers in patients with MS who where treated with probiotic supplementation [44]. A number of studies encourage the use of pre- and probiotics in patients with MS due to their benefits in maintaining homeostasis of the CNS, improving the intestinal microbial balance and regulating the composition of gut microbiota. In these studies the combination of prebiotics and probiotics is highly recommended [29].”

References

29. Riccio P, Rossano R. Nutrition facts in multiple sclerosis. ASN Neuro. 2015;7(1).

42. Moles L, Otaegui D. The Impact of Diet on Microbiota Evolution and Human Health. Is Diet an Adequate Tool for Microbiota Modulation? Nutrients. 2020;12(6).

43. Lombardi VC, De Meirleir KL, Subramanian K, Nourani SM, Dagda RK, Delaney SL, et al. Nutritional modulation of the intestinal microbiota; future opportunities for the prevention and treatment of neuroimmune and neuroinflammatory disease. J Nutr Biochem. 2018;61:1-16.

44. Kouchaki E, Tamtaji OR, Salami M, Bahmani F, Daneshvar Kakhaki R, Akbari E, et al. Clinical and metabolic response to probiotic supplementation in patients with multiple sclerosis: A randomized, double-blind, placebo-controlled trial. Clin Nutr. 2017;36(5):1245-9.

Reviewer 2 Report

This is a narrative review of a timely topic. The individual chapters a brief but sufficient to get orientation. I would like to see following adaptations and correction of false/premature statements:

  • Abstract/text: the current knowledge about the association of vitamin deficiency and MS pathogenesis is vague, this needs to be corrected.
  • Line 46: if a healthy diet ameliorates the disease course, further details are required.
  • The caveats of vitamin supplementations, particularly of fat-soluble vitamins is not covered sufficiently.
  • Line 311: The authors report about evidence for benefits of different diets in MS. This statement is incorrect., none of the studies evidence for mitigation of the MS course. Thus the entire chapters needs tob e rephrased.
  • Line 373: DMTs are not a prophylactic therapy.
  • Conclusion: the authors overestimate the subtle evidence on small aspects of symptoms or translate in-vitro/experimental findings in an incorrect fashion. This is indeed dangerous. It is clear that not larger trials are needed but basic research needs to show conclusively a hypothesis.

Author Response

Dear Reviewer,

On behalf of all co-authors, we are grateful for your fruitful comments upon our manuscript entitled “The role of diet and interventions on multiple sclerosis: A review”

We have the honor to submit our revised manuscript based on the remarks that have been proposed.

Sincerely yours

N. Grigoriadis

Q1. Abstract/text: the current knowledge about the association of vitamin deficiency and MS pathogenesis is vague, this needs to be corrected.

We thank Reviewer#2 for this comment.

In the abstract/text we rephrased, §8, Lines 88-89 as follows:

Studies report that low levels of vitamin D are associated with higher risk for development and relapse of MS [8, 17].”

References

  1. Dos Passos GR, Sato DK, Becker J, Fujihara K. Th17 Cells Pathways in Multiple Sclerosis and Neuromyelitis Optica Spectrum Disorders: Pathophysiological and Therapeutic Implications. Mediators Inflamm. 2016;2016:5314541.
  2. Pierrot-Deseilligny C, Souberbielle JC. Vitamin D and multiple sclerosis: An update. Mult Scler Relat Disord. 2017;14:35-45.

Also, we point out in our manuscript in §8, Lines 94-97 the following:

“However, Bagur et al reported in their systematic review, that existing studies on the effect of vitamin D supplementation in MS are inconsistent with respect to EDSS, MRI lesions, overall functional status, and relapse rate [13].”

References

13.Bagur MJ, Murcia MA, Jimenez-Monreal AM, Tur JA, Bibiloni MM, Alonso GL, et al. Influence of Diet in Multiple Sclerosis: A Systematic Review. Adv Nutr. 2017;8(3):463-72.

We also added in §9, Line 103:

Association between pathogenesis of MS and vitamin A remains undefined.”

Also, we report about the possible role between vitamin A and MS pathogenesis by quoting the following in §9, Lines 104-106:

 “Association between pathogenesis of MS and vitamin A remains undefined. Studies in animal models demonstrate a possible role of vitamin A in modulation of immunity [12, 22]. A negative correlation has been found between development of MS and low levels of vitamin A in plasma [23].”

References

  1. Miller ED, Dziedzic A, Saluk-Bijak J, Bijak M. A Review of Various Antioxidant Compounds and their Potential Utility as Complementary Therapy in Multiple Sclerosis. Nutrients. 2019;11(7).
  2. Dietary Reference Intakes for Vitamin A, Vitamin K, Arsenic, Boron, Chromium, Copper, Iodine, Iron, Manganese, Molybdenum, Nickel, Silicon, Vanadium, and Zinc. Washington (DC)2001.
  3. Xiao S, Jin H, Korn T, Liu SM, Oukka M, Lim B, et al. Retinoic acid increases Foxp3+ regulatory T cells and inhibits development of Th17 cells by enhancing TGF-beta-driven Smad3 signaling and inhibiting IL-6 and IL-23 receptor expression. J Immunol. 2008;181(4):2277-84.

- Q2. Line 46: if a healthy diet ameliorates the disease course, further details are required.

We thank Reviewer#2 for this comment.

We included  further details about the effects of a healthy, anti-inflammatory diet in  §33, lines 378-373 as follows:

Overall, following an antiinflammatory diet, in terms of intermittent fasting, seems to regulate inflammation and protect against oxidative damage and progression of MS in experimental models [29]. Choi et al report in their study that an anti-inflammatory diet that included mainly caloric restriction diet, showed promising results by reducing inflammation and promoting regeneration in animal models with experimented autoimmune encephalomyelitis [56].     

References

  1. Riccio P, Rossano R. Nutrition facts in multiple sclerosis. ASN Neuro. 2015;7(1).
  2. Choi IY, Piccio L, Childress P, Bollman B, Ghosh A, Brandhorst S, et al. A Diet Mimicking Fasting Promotes Regeneration and Reduces Autoimmunity and Multiple Sclerosis Symptoms. Cell Rep. 2016;15(10):2136-46.

- Q3.The caveats of vitamin supplementations, particularly of fat-soluble vitamins is not covered sufficiently.

We thank Reviewer#2 for this comment.

In the manuscript in §8, Lines 94-97 and we have proceeded with the following changes:

“It has been suggested that empirical replacement with high doses of vitamin D supplementation (at least 4000 IU/ day orally) and for a prolonged period appears to be safe and is associated with low risk for adverse events, although available data are limited [12,19-21].”

References

  1. Miller ED, Dziedzic A, Saluk-Bijak J, Bijak M. A Review of Various Antioxidant Compounds and their Potential Utility as Complementary Therapy in Multiple Sclerosis. Nutrients. 2019;11(7).
  2. Tredinnick AR, Probst YC. Evaluating the Effects of Dietary Interventions on Disease Progression and Symptoms of Adults with Multiple Sclerosis: An Umbrella Review. Adv Nutr. 2020;11(6):1603-15.
  3. Dobson R, Cock HR, Brex P, Giovannoni G. Vitamin D supplementation. Pract Neurol. 2018;18(1):35-42.
  4. Rito Y, Torre-Villalvazo I, Flores J, Rivas V, Corona T. Epigenetics in Multiple Sclerosis: Molecular Mechanisms and Dietary Intervention. Cent Nerv Syst Agents Med Chem. 2018;18(1):8-15.

We also rephrased  §9, Lines 106-109 as follows:

A randomized controlled trial showed benefits in fatigue, depression and cognitive status of MS patients supplemented with high doses of vitamin A (400 IU/day), which were considered safe and were not associated with adverse effects [24].

References

  1. Bitarafan S, Saboor-Yaraghi A, Sahraian MA, Soltani D, Nafissi S, Togha M, et al. Effect of Vitamin A Supplementation on fatigue and depression in Multiple Sclerosis patients: A Double-Blind Placebo-Controlled Clinical Trial. Iranian journal of allergy, asthma, and immunology. 2016;15(1):13-9.

- Q4. Line 311: The authors report about evidence for benefits of different diets in MS. This statement is incorrect., none of the studies evidence for mitigation of the MS course. Thus the entire chapters needs to be rephrased.

We thank Reviewer#2 for this comment.

We definitely agree we should better define the limits of the existing evidence on the potential role of the dietary interventions in MS course overall.

In the manuscript we rephrased the statements where we reported about evidence for benefits of diets in MS in §27, Lines 310-314 as follows:

  “ However, several studies provide at least some indications on the potential role of dietary habits in MS course. Among the most popular dietary interventions overall are the Mediterranean, The Paleolithic, the Swank, the McDougall and the Hyberbolic-caloric restriction diet [1, 32].

References

  1. Penesova A, Dean Z, Kollar B, Havranova A, Imrich R, Vlcek M, et al. Nutritional intervention as an essential part of multiple sclerosis treatment? Physiol Res. 2018;67(4):521-33.
  2. Altowaijri G, Fryman A, Yadav V. Dietary Interventions and Multiple Sclerosis. Curr Neurol Neurosci Rep. 2017;17(3):28.

And we rephrased in §28, Lines 323-324 as follows:

“Thus, it has been suggested that Mediterranean diet is associated with low risk for MS onset.”

-Q5: Line 373: DMTs are not a prophylactic therapy.

We thank Reviewer#2 for this comment.

We agree that the term “prophylactic” may not adequately correspond to the overall concept of disease modifying treatments in controlling MS activity. Definition of the use of DMTs is incorporated in the manuscript in the introduction section in §1, Lines 40-42 as follows:

Several disease modifying treatments (DMTs) are currently in use in order to manage the ongoing disease activity in an attempt to control relapses and disability progression [4].

References

  1. Koltuniuk A, Chojdak-Lukasiewicz J. Adherence to Therapy in Patients with Multiple Sclerosis-Review. Int J Environ Res Public Health. 2022;19(4).

We also rephrased §35, Lines 373-377 as follows:

Although a balanced diet involving high amount of fruit, vegetables and low fat may not replace DMTs in controlling disease activity in MS, it may have an add on value in a more efficient management of the disease overall. Existing evidence indicates that nutrition and diet may play a role in MS pathogenesis and course.

-Q6: The authors overestimate the subtle evidence on small aspects of symptoms or translate in-vitro/experimental findings in an incorrect fashion. This is indeed dangerous. It is clear that not larger trials are needed but basic research needs to show conclusively a hypothesis.

We thank Reviewer#2 for this comment. We probably did not make clear that our aim was to provide a comprehensive scientific evidence on the potential underlying mechanisms related to the impact of nutrients and diet on immune-mediated demyelination. In order to do so, we had to take advantage of key available data from in vitro, experimental and clinical studies. Of note, by no means we would provide any recommendation of clinical/practical value on the basis of the existing data, particularly those being reported in our manuscript. On the other hand, the existing data may not be circumvent and we strongly suggest large, well-scheduled clinical trials containing both clinical and biochemical, molecular, metagenomic and metabolomic technologies aiming to clarify the role of diet in MS management.

We consider our current position clearly defined in the conclusion session. 

In the conclusion of our manuscript §35, we replaced Lines 382-385 as follows:

“Furthermore, we strongly suggest large, well-scheduled clinical trials containing both clinical and biochemical, molecular, metagenomic and metabolomic technologies aiming to clarify the role of diet in MS management.”